# XSpecMesh: Quality-Preserving Auto-Regressive Mesh Generation Acceleration via Multi-Head Speculative Decoding

Dian Chen [*1]   Yansong Qu [*1]   Xinyang Li [1]   Ming Li [2]   Shengchuan Zhang [1]

## Abstract

Current auto-regressive models can generate high-quality, topologically precise meshes; however, they necessitate thousands—or even tens of thousands—of next-token predictions during inference, resulting in substantial latency. We introduce XSpecMesh, a quality-preserving acceleration method for auto-regressive mesh generation models. XSpecMesh employs a lightweight, multi-head speculative decoding scheme to predict multiple tokens in parallel within a single forward pass, thereby accelerating inference. We further propose a verification and resampling strategy: the backbone model verifies each predicted token and resamples any tokens that do not meet the quality criteria. In addition, we propose a distillation strategy that trains the lightweight decoding heads by distilling from the backbone model, encouraging their prediction distributions to align and improving the success rate of speculative predictions. Extensive experiments demonstrate that our method achieves a $1.7\times$ speedup without sacrificing generation quality. Our code will be released.

## 1. Introduction

Triangular meshes constitute the foundation of 3D representation and are extensively employed across industries, including virtual reality, gaming, animation, and product design. High-quality meshes exhibiting precise topology are essential for downstream tasks, such as mesh editing, skeletal rigging, texture mapping, and animation. However, constructing meshes with fine-grained topology remains a

---
*Equal contribution . Project lead: Yansong Qu. [1]Key Laboratory of Multimedia Trusted Perception and Efficient Computing, Ministry of Education of China, Xiamen University [2]Shandong Inspur Database Technology Co., Ltd.. Correspondence to: Shengchuan Zhang <zsc_2016@xmu.edu.cn>.

*Proceedings of the $43^{rd}$ International Conference on Machine Learning*, Seoul, South Korea. PMLR 306, 2026. Copyright 2026 by the author(s).

labor-intensive endeavor that requires substantial design effort, thus impeding the advancement of 3D content creation. Recent works employ auto-regressive architectures (Siddiqui et al., 2024; Chen et al., 2024a;b; Weng et al., 2025; Zhao et al., 2025a; Liu et al., 2025) for token-based mesh generation, they directly generate mesh vertices and faces while demonstrating the capacity to produce topologically precise meshes. However, the auto-regressive paradigm incurs high inference latency: existing auto-regressive mesh generation models depend on next-token predictions, requiring thousands to tens of thousands of forward passes to produce a single 3D mesh.

We draw inspiration from Speculative Decoding (Leviathan et al., 2023; Chen et al., 2023) in efficient LLM inference, which typically employs a draft model with significantly fewer parameters than the original. The draft model generates candidate tokens, which the original model then verifies—enabling near-draft-model generation speed while preserving the original model's generation quality. However, draft models must satisfy stringent criteria: their parameter count must be sufficiently constrained to facilitate accelerated inference, and their predictions must closely align with the distribution of the original model. Consequently, deriving such draft models remains a significant challenge(Chen et al., 2023; Leviathan et al., 2023). On the other hand, we note that, unlike auto-regressive language models which frequently employ larger vocabularies to enhance expressiveness (Tao et al., 2024; Huang et al., 2025), existing auto-regressive mesh generation models typically utilize efficient, compressed representations to minimize vocabulary size. Table 1 summarizes these disparities. This discrepancy motivates us to explore a more lightweight decoding design to obtain the probability distribution over the vocabulary.

To this end, we introduce XSpecMesh, a novel framework that accelerates auto-regressive mesh generation models while preserving generation quality. The framework implements multi-head speculative decoding to accelerate inference: multiple lightweight decoding heads simultaneously predict a sequence of subsequent tokens in a single forward pass. These decoding heads leverage cross-attention mechanisms with the generation conditions to enhance prediction accuracy. Furthermore, we introduce a verification and re-

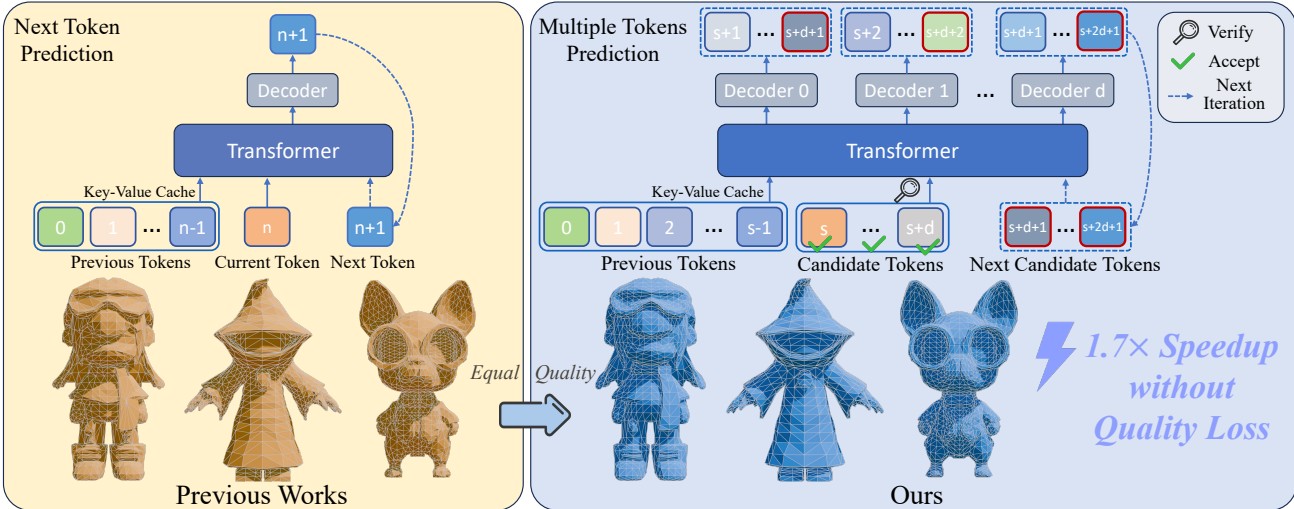

*Figure 1.* **The differences between our framework and previous works.** We propose XSpecMesh, a method for accelerating auto-regressive mesh generation models via multi-head speculative decoding, instead of relying on traditional next-token prediction. In a single forward pass, multiple decoding heads predict several tokens, verify the candidate tokens, and resample candidate tokens for the next iteration. Our approach delivers a $1.7\times$ speedup while preserving generation quality.

sampling strategy to evaluate candidate tokens predicted by the decoding heads, resampling those that fail to meet quality criteria, thereby ensuring that output quality remains uncompromised. Finally, we employ backbone distillation training to encourage the decoding heads' predictive distributions to approximate that of the backbone model, allowing the backbone to accept their predictions. Figure 1 illustrates the differences between our framework and previous works.

We present XSpecMesh, a method for accelerating inference in auto-regressive mesh generation models without sacrificing generation quality. Our contributions can be summarized as follows:

- We propose XSpecMesh, a method to accelerate auto-regressive mesh generation models without compromising generation quality, by employing multiple cross-attention speculative decoding heads for multi-token prediction.

- We develop a verification and resampling strategy that, within a single forward pass, employs the backbone model to verify candidate tokens and resample those that do not meet predefined quality criteria, thereby ensuring uncompromised generation quality.

- We further introduce a distillation strategy to train decoding heads, aligning their prediction distribution with the backbone model to improve the success rate of speculative predictions.

- Extensive experiments demonstrate that our method significantly accelerates inference without sacrificing generation quality, achieving a $1.7\times$ speedup.

## 2. Related Works

### 2.1. 3D Mesh Generation

Due to the complexity of direct mesh generation, many 3D synthesis methods utilize intermediate representations—such as voxels (Wu et al., 2016), point clouds (Luo & Hu, 2021; Jun & Nichol, 2023; Qi et al., 2017; Shen et al., 2025c), implicit fields (Chen & Zhang, 2019; Park et al., 2019), or 3DGS(Kerbl et al., 2023; Li et al., 2024; Wang et al., 2024; **?**; 2026; Shen et al., 2025b; Qu et al., 2024; Dai et al., 2025; Qu et al., 2025a)—to avoid modeling meshes directly. Representative approaches include optimizing 3D representations within pretrained 2D diffusion models via score-distillation sampling (SDS) (Poole et al., 2022; Wang et al., 2023; Zhu et al., 2023; Li et al., 2023; Lin et al., 2023; Tang et al., 2023; Yi et al., 2024; Li et al., 2024); generating multi-view-consistent images with 2D diffusion models and reconstructing meshes from them (Liu et al., 2023; Shi et al., 2023; Qu et al., 2025b); 3D transformer models (Hong et al., 2023; Tang et al., 2024a; Xu et al., 2024), and the recent 3D latent diffusion models (Zhang et al., 2024; Xiang et al., 2025; Hunyuan3D et al., 2025; Wu et al., 2024; Zhao et al., 2025b) that achieve high-quality shape generation. These approaches typically apply Marching Cubes (Lorensen & Cline, 1998) in post-processing to extract meshes, frequently introducing topological artifacts. In contrast, MeshGPT (Siddiqui et al., 2024), which integrates VQ-VAE (Van Den Oord et al., 2017) with a transformer (Vaswani et al., 2017) for auto-regressive mesh generation, produces high-quality topological meshes; however, it is confined to low-polygon meshes and single-category shapes. A subsequent series of auto-regressive mesh generation

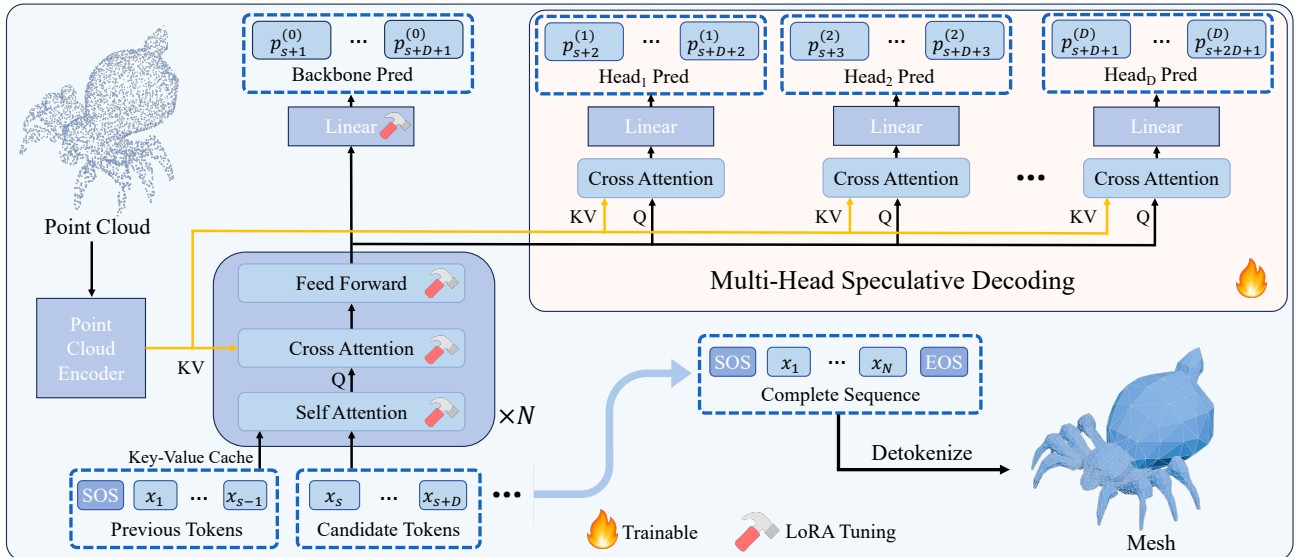

*Figure 2.* **Overview of our method.** Left: A pretrained transformer-based auto-regressive mesh generation model, fine-tuned with LoRA. Top-right: The transformer's final hidden layer is decoded by $D$ cross-attention decoding heads, the $d$-th head predicts the $(d+1)$-th next token. Bottom-right: The complete generated token sequence is detokenized to produce the mesh.

methods (Chen et al., 2024b;c; Tang et al., 2024b; Hao et al., 2024; Chen et al., 2024a; Liu et al., 2025), has demonstrated the ability to synthesize topologically precise meshes, BPT (Weng et al., 2025) and DeepMesh (Zhao et al., 2025a) further scale auto-regressive mesh generation to large datasets through efficient tokenization schemes. However, the intrinsic latency of the auto-regressive paradigm hinders its applicability. In this paper, we therefore propose a novel method to accelerate auto-regressive mesh generation while preserving generation quality.

### 2.2. Acceleration of Auto-Regressive Model

Various strategies have been proposed to accelerate auto-regressive language models: weight pruning methods (Frantar & Alistarh, 2023; Sanh et al., 2020) eliminate redundant parameters to decrease computational load; quantization techniques (Frantar et al., 2022; Xiao et al., 2023) convert models into low-bit representations to cut memory and compute overhead; and sparsity-based approaches (Fedus et al., 2022; Fu et al.) reduce activation computations to improve efficiency. Nonetheless, these methods retain the conventional auto-regressive, token-by-token decoding paradigm. An alternative research direction (Gloeckle et al., 2024; Fan et al., 2025; Cai et al., 2024; Wang et al., 2025) attempts to predict multiple tokens in a single forward pass to reduce iterative decoding steps. The Speculative Decoding approaches (Chen et al., 2023; Sun et al., 2023; Leviathan et al., 2023) employ a draft model to generate tokens rapidly, then verify them with the original model to preserve generation quality. Certain efforts target acceleration of auto-regressive image synthesis: SJD (Teng et al., 2024) inte-

grates Speculative Decoding with Jacobi decoding, whereas ZipAR (He et al., 2024) exploits local sparsity for parallel token generation. To date, these acceleration studies have focused predominantly on language and image generation domains, with auto-regressive mesh generation remaining insufficiently explored.

## 3. Preliminary

### 3.1. Auto-Regressive Mesh Generation

An auto-regressive mesh generation framework comprises three fundamental components: a discrete mesh serialization method (Chen et al., 2024b; Weng et al., 2025) that converts vertices and faces into a token sequence; a transformer-based auto-regressive generator that, conditioned on input prompts, sequentially predicts each subsequent token to generate the token sequence; a deserialization method that reconstructs the 3D mesh vertices and faces from the generated sequence.

Auto-regressive models employ causal masking during training, so that, for a given sequence $x_{0:n}$, the model can perform, in a single forward pass, simultaneous computations of the predictive distributions for positions $1, 2, \ldots, n+1$:

$$p_1(x|x_0), \ p_2(x|x_{0:1}), \ \ldots, \ p_{n+1}(x|x_{0:n}). \quad (1)$$

For each position $i$, with corresponding target label $y_i$, the model is trained by minimizing the cross-entropy loss:

$$\mathcal{L} = \sum_i -\log p_i(y_i). \quad (2)$$

This property also means that, at inference time, by evaluating $p_{i+1}(x|x_{0:i})$, one can determine whether a candidate token $x_{i+1}$ aligns with the model's learned distribution. Our method leverages this property to accelerate generation without compromising quality.

# 4. Method

Our method aims to accelerate auto-regressive mesh generation models without compromising generation quality. We propose multi-head speculative decoding, in which multiple lightweight cross-attention decoding heads concurrently predict subsequent tokens, thereby accelerating the sequence generation process (Sec 4.1). Since these decoding heads' predictions may be imprecise, we employ the backbone model's robust prior to verify outputs—rejecting and resampling at the first invalid token—to guarantee generation quality (Sec 4.2). To enhance acceptance of decoding heads' proposals, we distill backbone knowledge into these heads during training, aligning their output distributions with the backbone's (Sec 4.3). Figure 2 provides an overview of our method.

## 4.1. Multi-Head Speculative Decoding

Auto-regressive models exhibit excellent generation quality, however, their inference relies on sequential, token-by-token generation, leading to high latency. To alleviate this bottleneck, we introduce multi-head speculative decoding. In auto-regressive mesh generation models, the vocabulary size is considerably smaller than that of LLMs (Table 1), resulting in a relatively simple decoding process. Therefore, we propose a more efficient approach that employs multiple lightweight decoding heads to process the transformer's final hidden layer and predict subsequent tokens.

Specifically, the backbone model comprises $N$ transformer blocks, each containing: a self-attention layer, a cross-attention layer for injecting the generation condition $c$, and a feed-forward network. Let $s$ denote the current sequence position, and assume tokens $x_0$ through $x_{s-1}$ are stored in the key–value cache. Denote the layer-0 hidden state as $h_s^0 = x_s$. Then, for $l = 0, 1, \ldots, N - 1$, the $(l + 1)$-th hidden state is computed as $h_s^{l+1} = \text{block}^l(h_s^l, c)$. Define the final hidden state as $h_s = h_s^N$. The backbone model subsequently decodes $h_s$ through a linear layer $W^{(0)}$ to yield the probability distribution for the next token at position $s + 1$:

$$p_{s+1}^{(0)} = \text{softmax}(W^{(0)} \cdot h_s). \qquad (3)$$

Given the generation condition $c$, we employ multiple cross-attention decoding heads to decode $h_s$, with the $d$-th decod-

| Method | BPT | DeepMesh | LLaMa 3 | Qwen3 |
|---|---|---|---|---|
| Vocab Size | 5120 | 4736 | 128K | 152K |

*Table 1.* **The difference in vocabulary size between auto-regressive mesh generation models and language models.** Language models (Grattafiori et al., 2024; Yang et al., 2025) tend to use larger vocabularies to enhance expressiveness, whereas auto-regressive mesh generation models favor efficient compressed representations to reduce vocabulary size.

ing head predicting the token at position $s + d + 1$:

$$p_{s+d+1}^{(d)} = \text{softmax}(W^{(d)} \cdot \text{CrossAttn}^{(d)}(h_s, c)). \qquad (4)$$

Compared to decoding via an MLP, using a cross-attention mechanism allows the decoding heads to better align with the input conditional features, thereby improving the accuracy of subsequent-token predictions. Finally, we sample from probability distributions $p_{s+1}^{(0)}, p_{s+2}^{(1)}, \ldots, p_{s+D+1}^{(D)}$ to generate the tokens $x_{s+1}, x_{s+2}, \ldots, x_{s+D+1}$.

## 4.2. Verification and Resampling

After generating the next $D + 1$ tokens from position $s$ via the backbone model and $D$ decoding heads, the straightforward approach is to append these tokens to the existing sequence and resume prediction at position $s + D + 1$. However, due to potential inaccuracies of the decoding heads, this strategy can drastically degrade the generated sequence's quality. We therefore propose a verification strategy that leverages the backbone model to simultaneously verify and resample tokens in a single forward pass.

Specifically, we leverage the backbone model's prior judgment to determine whether to accept tokens predicted by the decoding heads. Let $s$ denote the current accepted sequence position. In a single forward pass, the backbone model employs causal masking on the sequence $x_{s:s+D}$ to obtain $p_{s+1:s+D+1}^{(0)}$, and based on this probability distribution, partially accepts a prefix $x_{s:s^*-1}$. Subsequently, with the backbone model and $D$ decoding heads, we resample tokens at positions $s^*$ to $s^* + D$. We apply a probability-threshold-based criterion: a token $x_i$ is accepted if $p_i^{(0)}(x_i) > \delta$. Figure 3 provides a detailed illustration of this process.

By verifying with the backbone model and sampling with multiple decoding heads, we reduce the number of forward passes through the backbone model while preserving generation quality, thus speeding up the overall generation process. Algorithm 1 presents a detailed description of the multi-head speculative decoding procedure.

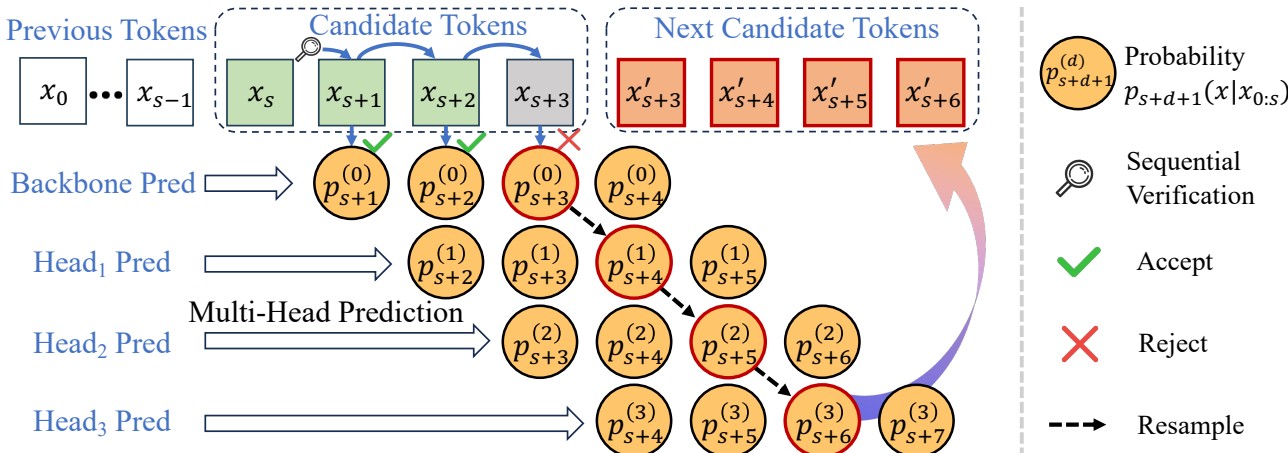

**Figure 3. Verification and resampling.** The figure uses $D = 3$ as an example to illustrate the process. Each candidate token sampled in a forward pass must be verified by the backbone model: if $p_i^{(0)}(x_i) > \delta$, token $x_i$ is accepted and verification proceeds to the next token, until the first token $x_{i'}$ that fails the verification condition. Then resample a token at position $i'$, forming the candidate tokens for the next iteration.

### 4.3. Backbone Distillation Training

Analogous to Speculative Decoding, in which the draft model's output distribution must closely match that of the original model, our framework requires the decoding heads' output distributions to align with the backbone model's distribution to ensure acceptance of their predictions. To this end, we distill the backbone model to train decoding heads. We sample point clouds from the dataset and employ the backbone model to generate sequences, which serves as the ground truth labels $y_{0:n}$ for decoding heads training, We train the $d$-th decoding head using the cross-entropy loss:

$$\mathcal{L}_d = \sum_s -\log p_{s+d+1}^{(d)}(y_{s+d+1}). \tag{5}$$

With increasing $d$, the accuracy of the $d$-th decoding head declines, potentially causing gradient instability. To mitigate this issue, we introduce a weighting function $w(d)$, which decreases as $d$ increases. Accordingly, the overall loss for the $D$ decoding heads is formulated as follows:

$$\mathcal{L}_{\text{mhd}} = \sum_{d=1}^{D} w(d) \cdot \mathcal{L}_d. \tag{6}$$

Following decoding heads training, they are deployed for inference acceleration. Empirical evaluation, however, indicates that the speed-up benefits are modest. This limitation arises because the backbone model is optimized under a next-token prediction paradigm, making direct decoding of subsequent tokens from the hidden state $h_s$ infeasible. To mitigate this issue, we fine-tune the backbone model's linear layer via LoRA (Hu et al., 2022), enabling the decoding heads to more effectively derive multiple subsequent

token predictions from $h_s$. Training proceeds in two stages. In the first stage, we train only the decoding heads while freezing the backbone model to prevent unstable gradients from the decoding heads in the early training stage from affecting the backbone model. In the second stage, we jointly train both the decoding heads and LoRA. Furthermore, we integrate the backbone model's prediction loss $\mathcal{L}_{\text{backbone}} = \sum_s -\log p_{s+1}^{(0)}(y_{s+1})$ into the overall objective with a substantial weighting factor $\lambda$, ensuring gradients from the decoding heads do not diverge the backbone distribution from its original form. The loss function for the second stage is formulated as follows:

$$\mathcal{L}_{\text{total}} = \lambda \mathcal{L}_{\text{backbone}} + \mathcal{L}_{\text{mhd}}. \tag{7}$$

Although during training LoRA introduces two low-rank matrices $A$ and $B$ to each original linear layer weight matrix $W$, at inference time these LoRA weights can be merged with the original weights $W_{\text{origin}}$ via a simple preprocessing step to form the merged weight $W_{\text{merge}} = W_{\text{origin}} + AB$. Therefore, introducing LoRA incurs no additional computational overhead. Upon fine-tuning the backbone model with LoRA, the decoding heads are able to accurately predict subsequent tokens, significantly increasing the decoding speed.

## 5. Experiments

### 5.1. Experiment Settings

**Serialization strategy.** We adopt the same serialization strategy as BPT(Weng et al., 2025), which is an irregular compressed mesh serialization method. Specifically, BPT hierarchically encodes mesh vertices using block indices and offset indices to balance vocabulary size and sequence

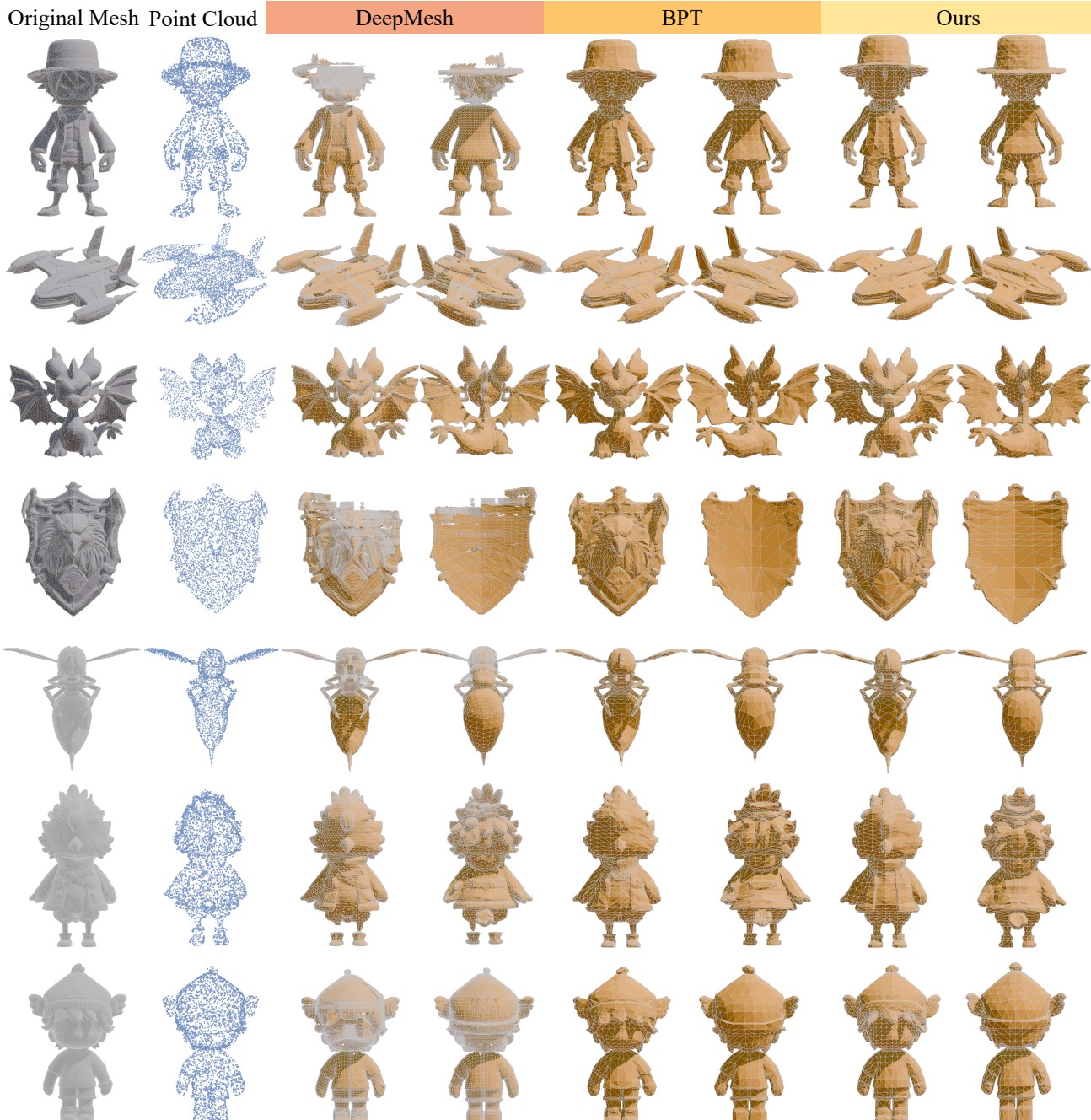

*Figure 4.* **Comparison of our method with the base model BPT and another mesh generation model DeepMesh.** Our acceleration method, built upon BPT, substantially accelerates generation while preserving BPT's shape and topological fidelity.

length, and further applies patchify operations to compress the substantial redundant repetition of vertices in the sequence. Our method can effectively accelerate inference under such irregular compressed sequences.

**Implementation Details.** We adopt BPT as our base model: an auto-regressive mesh generation model pretrained on a large-scale, high-quality dataset. We train on a subset of Objaverse (Deitke et al., 2023) containing approximately 10K shapes. In the first stage, we train only the decoding heads, setting the loss weight for the $d$-th decoding head to $w(d) = 0.8^d$. In the second stage, we jointly train the LoRA adapters and the decoding heads; to prevent the backbone model's distribution from drifting, we assign a relatively large weight $\lambda = 50$ to the backbone loss. See the Appendix for more details.

**Algorithm 1** Multi-Head Speculative Decoding

**Input**: Condition $c$, Backbone Model $\mathcal{M}$, Multi-Head Speculative Decoder $\{\mathcal{H}_i\}_{i=1}^{D}$

**Output**: Mesh Sequence $x_{0:i_{\text{EOS}}}$

1: Let $x_0 \leftarrow \text{SOS}$, $x_{1:D} \sim U(0, V)$, $s \leftarrow 0$.
2: **while** $s < L_{max}$ and $x_{0:s} \neq \text{EOS}$ **do**
3:     $p_{s+1:s+D+1}^{(0)}$, $h_{s:s+D} \leftarrow \mathcal{M}(x_{s:s+D}, c)$
    {forward with causal mask}
4:     **for** $i = 1$ to $D$ **do**
5:         $p_{s+1+i:s+D+1+i}^{(i)} \leftarrow \mathcal{H}_i(h_{s:s+D}, c)$
6:     **end for**
7:     $s^* \leftarrow s + 1$
8:     **while** $s^* < s + D + 1$ and $p_{s^*}^{(0)}(x_{s^*}) > \delta$ **do**
9:         $s^* \leftarrow s^* + 1$ {verify and accept}
10:     **end while**
11:     $x_{s^*:s^*+D} \leftarrow \text{sample}(p_{s^*}^{(0)}, p_{s^*+1}^{(1)}, \dots, p_{s^*+D}^{(D)})$
    {resample from the first rejected position $s^*$}
12:     $s \leftarrow s^*$
13: **end while**
14: **return** $x_{0:i_{\text{EOS}}}$

| Method | CD ↓ | HD ↓ | US ↑ | Avg. Lat. ↓ |
|---|---|---|---|---|
| DeepMesh* | 0.1323 | 0.2648 | 27% | 979.6s |
| BPT | 0.1165 | 0.2223 | 37% | 257.6s |
| Ours | 0.1168 | 0.2261 | 36% | **151.4s** |

*Table 2.* **Quantitative comparison with other methods.** Our approach achieves generation quality comparable to the base model BPT while delivering significantly faster generation speed than BPT. Avg. Lat. denotes the average latency to generate the complete mesh sequence (measured on the RTX 3090). DeepMesh* was tested using its 0.5B version.

**Baselines.** We compare our method against the base model BPT and another state-of-the-art auto-regressive mesh generation model, DeepMesh (Zhao et al., 2025a). Since DeepMesh has only released a 0.5B-parameter configuration, we use this version for evaluation.

**Metrics.** We follow the evaluation procedure of previous work (Weng et al., 2025; Zhao et al., 2025a; Liu et al., 2025), and generate 200 test meshes via the generation model (Xiang et al., 2025; Zhao et al., 2025b) (see the Appendix for more details). We uniformly sample 1,024 points from the surfaces of ground-truth and generated meshes, computing Chamfer Distance (CD) and Hausdorff Distance (HD) as objective quality metrics. Additionally, a user study (US) is conducted to capture subjective assessments. For speedup evaluation, we follow the methodology of previous work (Fu et al., 2024; Xia et al., 2022; Chen et al., 2023) and define the Step Compression Ratio as: $\text{SCR} = \frac{\text{number of generated and accepted tokens}}{\text{number of decoding steps}}$, where a decoding step denotes the process of verifying and decoding multiple tokens in a single forward pass. Since we introduced additional decoding heads, we measured the latency of a single decoding step (Step Latency) on an RTX 3090. Finally, we computed the actual speedup ratio (Speedup) based on SCR and Step Latency.

### 5.2. Qualitative Results

We perform a qualitative comparison of our method against established baselines, presenting several challenging examples in Figure 4. Although DeepMesh can generate higher-resolution meshes, its truncated-window training induces

context loss, resulting in fragmented meshes. In contrast, BPT yields more consistent generation results, while our approach achieves shape and topological fidelity comparable to BPT.

### 5.3. Quantitative Results

Table 2 summarizes the results of our quantitative comparison. DeepMesh is capable of generating high-resolution meshes, which has earned it a certain level of popularity in user study. However, owing to its propensity to produce fragmented and incomplete meshes, DeepMesh exhibits higher CD and HD values. By contrast, the results generated by BPT demonstrate greater consistency. Since our method produces results highly similar to BPT, the corresponding CD and HD metrics are comparable. Moreover, in the user study where methods were anonymized, participants were unable to differentiate between our method's outputs and those of BPT, yielding comparable survey scores. Overall, our method matches the baseline BPT in generation quality while significantly reducing complete mesh sequence generation latency.

### 5.4. Ablation Study

**Decoding head architectures and training strategies.** We first compared the quality of the generated shapes and the achieved speed-up under different decoding head architectures and training strategies: A. Baseline model: BPT; B. MLP decoding heads, training only the first-stage decoding heads; C. MLP decoding heads, first training the first-stage decoding heads, then jointly training LoRA adapters and decoding heads in a second stage; D. Cross-attention decoding heads, training only the first-stage decoding heads; E. Cross-attention decoding heads, first training the first-stage decoding heads, then jointly training LoRA adapters and decoding heads in a second stage.

Table 3 presents the evaluation results for different configurations. Compared to the MLP decoding heads, the cross-attention decoding heads, despite incurring higher step latency, more effectively integrate conditional information into the generation process, thereby yielding more accurate

| Configuration | | CD ↓ | HD ↓ | SCR ↑ | Step Latency ↓ | Speedup ↑ |
|---|---|---|---|---|---|---|
| **A** | *w.* BPT | 0.1165 | 0.2223 | 1.000 | 40.51ms | 1.00× |
| **B** | *w.* MLP Decoder | 0.1195 | 0.2241 | 1.181 | 44.89ms | 1.07× |
| **C** | *w.* MLP Decoder & LoRA | 0.1267 | 0.2485 | 1.909 | 44.92ms | 1.65× |
| **D** | *w.* CA Decoder | 0.1167 | 0.2229 | 1.334 | 47.81ms | 1.13× |
| **E** | *w.* CA Decoder & LoRA (Ours) | 0.1168 | 0.2261 | **2.021** | 47.83ms | **1.71×** |

*Table 3.* **Ablation across different configurations.** We compare MLP decoding heads versus Cross-Attention (CA) decoding heads, and evaluate the effect of two-stage LoRA joint training with the decoding heads. The Cross-Attention decoding heads incorporate generation conditions, achieving excellent performance in both generation quality and speedup.

predictions of subsequent tokens and consequently improving the SCR. After two-stage joint training with LoRA, the MLP decoding heads also achieve a comparably high speedup; however, their generation quality deteriorates to some extent. This degradation stems mainly from (1) Joint training with LoRA aligns the prediction distributions of the decoding heads with those of the backbone model, thereby increasing the backbone's propensity to accept the decoding head's outputs, and (2) the MLP decoding heads' predictions, lacking injected conditional information, produce some inaccuracies that the backbone model still accepts, thereby compromising overall quality. In contrast, integrating cross-attention decoding heads with LoRA joint training better aligns multi-token predictions with generation condition, resulting in superior performance in both generation quality and speedup ratio.

**Number of decoding heads.** Increasing the number of decoding heads raises SCR but also increases step latency. As shown in Figure 5(a), we present SCR and step latency for various numbers of decoding heads and subsequently compute speedup. At $D = 4$, speedup peaks at $1.71\times$.

**Verification criterion.** We use a threshold $\delta$ as the acceptance condition: a token $x_i$ is accepted if $p_i^{(0)}(x_i) > \delta$. As the hyperparameter $\delta$ increases, the criterion becomes stricter, leading to lower speedup but improved generation quality. Figure 5(b) illustrates the impact of varying $\delta$ on speedup, CD, and HD. At $\delta = 0.5$, our method achieves an optimal trade-off between speedup and generation quality, delivering substantial acceleration while preserving quality comparable to the baseline model. Furthermore, we compare two acceptance criteria—Probability-Threshold Acceptance and Top-$K_a$ Acceptance (a token $x_i$ is accepted if $x_i$ is among the top-$K_a$ tokens of $p_i^{(0)}$)—and present the results in Figure 6. For top-$K_a$ acceptance, a long-tail effect arises: certain candidate tokens within the top-$K_a$ may exhibit exceedingly low probabilities yet be accepted, severely degrading generation quality. Only for $K_a = 1$ does the model generate a reasonable shape. By contrast, probability-threshold acceptance demonstrates greater stability, yielding satisfactory results for thresholds between 0.1 and 0.5.

**Sampling strategies.** We compare two sampling strategies: Independent Sampling and Top-$K_s$ Probability-Tree Sampling, see the Appendix for details.

### 5.5. Watertightness and Manifoldness

Watertightness and manifoldness are important metrics for assessing whether generated meshes can be reliably used in downstream applications. To further verify that our method improves generation efficiency without sacrificing geometric validity, we conduct additional evaluations on watertightness and manifoldness for both existing autoregressive mesh generation methods and our proposed method. These evaluations complement standard generation-quality metrics by focusing on the topological and structural validity of the generated meshes, rather than only measuring their visual or distributional quality. Specifically, watertightness reflects whether a generated mesh forms a closed surface without holes or boundary gaps, while manifoldness indicates whether the local connectivity around vertices and edges satisfies valid surface constraints. Detailed quantitative results, evaluation protocols, and further analysis are provided in the Appendix. It is worth noting that autoregressive mesh generation is still at an early stage, and most existing methods primarily focus on producing plausible geometry rather than explicitly enforcing watertightness or manifoldness during generation. As a result, current methods still struggle to consistently produce outputs that are both watertight and manifold, suggesting that improving topological validity remains an important direction for future work.

### 6. Discussion

**Limitation.** Although our method accelerates the base model's generation speed without sacrificing output quality, it still relies on the base model as the backbone to verify candidate tokens and preserve generation quality. As a result, the overall performance of our approach remains tied to the capability and efficiency of the underlying base model. Nevertheless, this design allows our method to inherit the strengths of the backbone while providing consistent accel-

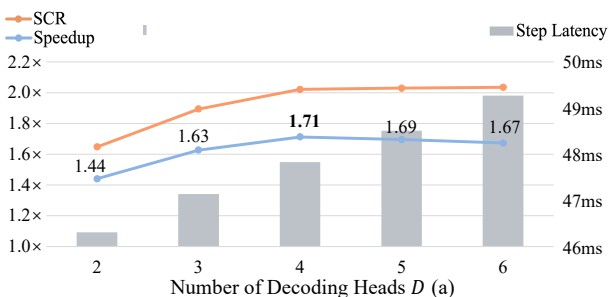
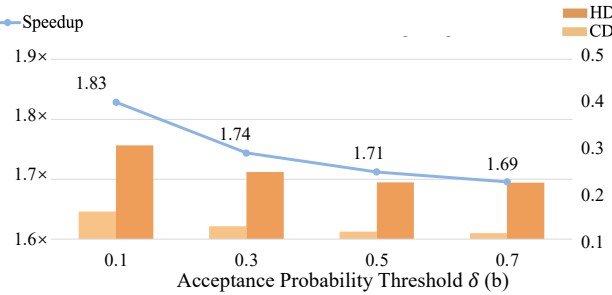

*Figure 5.* Left: ablation of the number of decoding heads $D$; speedup peaks at $D = 4$. Right: ablation of the acceptance probability threshold $\delta$; at $\delta = 0.5$, generation quality matches the base model while speedup exceeds $1.7\times$.

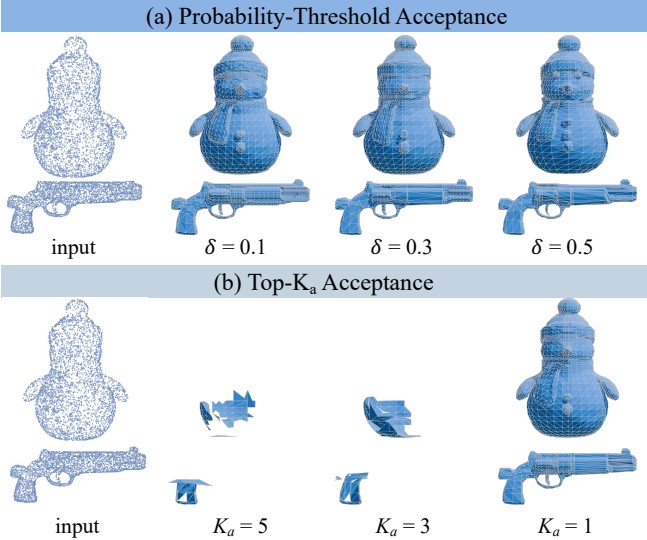

*Figure 6.* **Comparison of Probability-Threshold and Top-$K_a$ Acceptance.** Probability-threshold acceptance is more stable, generating reasonable shapes across thresholds.

eration in practice.

**Acceleration Bottlenecks.** Although our method consistently accelerates autoregressive mesh generation across different settings, the achieved speedup is still lower than the $2\times$–$3\times$ gains commonly reported for speculative decoding in large language models. We attribute this gap to several characteristics specific to autoregressive mesh generation. First, existing autoregressive mesh generation models are typically much smaller than modern LLMs. As a result, they contain less parameter redundancy, which limits the potential benefit of using a lightweight draft model to approximate the target model. Second, mesh token sequences often exhibit stronger local and structural dependencies than natural language sequences. A single geometric element, such as a vertex coordinate, face index, or local connectivity pattern, may be serialized into multiple tightly coupled tokens, so an error in one token can easily affect the validity of subsequent tokens. This makes accurate multi-token

drafting and acceptance more challenging. Third, recent mesh tokenization and serialization strategies are usually designed to be compact, reducing unnecessary sequence length and repeated patterns. While such compact representations improve generation efficiency in the standard autoregressive setting, they also leave less sequence-level redundancy for speculative decoding to exploit. Consequently, the attainable acceleration is naturally more limited than in LLM generation. Nevertheless, our method still provides consistent and practical speedups without degrading mesh generation quality, suggesting that speculative acceleration remains a promising direction for efficient autoregressive mesh generation.

## 7. Conclusion

We propose XSpecMesh, a speculative decoding framework for accelerating auto-regressive mesh generation models through multiple cross-attention decoding heads for multi-token prediction. By leveraging multi-head speculative decoding together with a verification and resampling strategy, our method improves inference efficiency while preserving generation quality. Extensive experiments show that XSpecMesh achieves up to a $1.7\times$ speedup over the base model without compromising output quality.

## Acknowledgements

This work was supported by the National Science and Technology Major Project (No. 2025YFE0113500), the National Science Fund for Distinguished Young Scholars (No. 62525605), and the National Natural Science Foundation of China (No. U25B2066, No. U22B2051, and No. 62272401).

## Impact Statement

This paper presents work whose goal is to advance the field of Machine Learning. There are many potential societal consequences of our work, none which we feel must be specifically highlighted here.

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

# A. Implementation Details

Training was performed on two NVIDIA A800 GPUs and took approximately eight hours. We used AdamW as the optimizer ($\beta_1 = 0.9, \beta_2 = 0.99$). To improve training stability, we applied global norm clipping to the gradients, limiting their overall norm to within 1.0. The training procedure comprised two stages. During stage one, we trained only the decoding head, employing a cosine learning rate schedule decaying from $5 \times 10^{-4}$ to $5 \times 10^{-5}$ over 30 epochs. Subsequently, we applied LoRA to fine-tune the backbone, jointly training both modules for 10 epochs with a cosine learning rate schedule decaying from $1 \times 10^{-4}$ to $1 \times 10^{-5}$. We set the LoRA rank to 16 and alpha to 32.

**Test set composition.** The composition of the test set is the same as in previous methods. We use image samples provided in the code repositories of Hunyuan3D 2.0 and Trellis as inputs, and generate 200 meshes as test examples to ensure that these data are unseen during training.

**Input point clouds.** The input point clouds are typically obtained either from scanned data or by sampling from the surfaces of meshes generated by diffusion-based models such as Trellis.

# B. Test Dataset

Our test data was generated from the generation model (Xiang et al., 2025; Zhao et al., 2025b) and covers a rich and diverse set of shapes. Moreover, we categorized the shapes in the test dataset into three different difficulty levels: level-0, level-1, and level-2. (1) level-0: Simple shapes with minimal detail. (2) level-1: Relatively complex shapes with a certain amount of detail. (3) level-2: Challenging shapes featuring a rich array of details. In the entire test dataset, level-0 accounts for approximately 20%, level-1 for 40%, and level-2 for another 40%. We showcase a subset of the shapes from the test set in Figure 7.

# C. Ablation of Sampling Strategies

We conducted a study of sampling strategies by comparing two methods: Independent Sampling (IS) and Top-$K_s$ Probability Tree Sampling (PTS).

**Independent Sampling.** Samples are drawn independently from each decoding head's probability distribution $p^{(d)}$, with token probabilities serving as sampling weights.

**Top-$K_s$ Probability Tree Sampling.** For each layer's distribution $p^{(d)}$, the Top-$K_s$ tokens by probability are selected to recursively construct a probability tree. Denote the probabilities of the Top-$K_s$ tokens at layer $d$ by $\{m_{i_{d,k}}^{(d)}\}_{k=1}^{K_s}$. The weight of a path from the root to a leaf is then computed as $\prod_{d=1}^{D} m_{i_{d,k}}^{(d)}$. To constrain tree-construction complex-

| Method | SCR ↑ | Step Latency ↓ | Speedup ↑ |
|---|---|---|---|
| IS | 2.021 | 47.83ms | 1.71× |
| PTS($K_s = 2$) | 2.030 | 48.06ms | 1.71× |
| PTS($K_s = 3$) | 2.033 | 48.47ms | 1.69× |
| PTS($K_s = 4$) | 2.036 | 49.08ms | 1.68× |

*Table 4.* **Comparison of Independent Sampling (IS) and Top-$K_s$ Probability Tree Sampling (PTS).** Top-$K_s$ PTS achieves a higher SCR, but due to the overhead of building the search tree at each iteration, its actual speedup is slightly lower than that of Independent Sampling.

| Method | CD ↓ | HD ↓ | Avg. Lat. ↓ |
|---|---|---|---|
| DeepMesh* | 0.1323 | 0.2648 | 979.6s |
| BPT | 0.1165 | 0.2223 | 257.6s |
| MeshSilk | 0.1111 | 0.2293 | 294.8 |
| MeshAnyV2 | 0.1708 | 0.3574 | 147.9s |
| TreeMeshGPT | 0.1871 | 0.3425 | 239.3s |
| Ours | 0.1168 | 0.2261 | 151.4s |

*Table 5.* **Broader comparison with other methods.**

ity, branches with cumulative weights below $1 \times 10^{-5}$ are pruned. Complete paths are then sampled according to their accumulated path probabilities.

Compared to IS, Top-$K_s$ PTS improves the step compression ratio (SCR) by considering combinations among sampled tokens, but because each iteration requires building a search tree—incurring additional overhead—it does not achieve a higher speedup. The results are shown in Table 4.

# D. User Study

We randomly selected 70 participants to complete a questionnaire as a subjective metric. Each questionnaire comprised 20 cases, resulting in 1,400 responses in total. Outputs from DeepMesh, BPT, and our method were randomly shuffled and anonymized to ensure fairness. For each case, participants were instructed to holistically evaluate both the generated shape and wireframe topology, then select the most favorable result. Owing to its tendency to generate fragmented and incomplete meshes, DeepMesh received relatively fewer votes. By contrast, participants struggled to distinguish between BPT and our method, resulting in nearly identical vote counts for these two approaches.

# E. Analysis of Qualitative and Quantitative Comparisons

While DeepMesh can produce meshes with greater face counts and finer details, it requires substantially longer token sequences. To mitigate this, DeepMesh was trained with

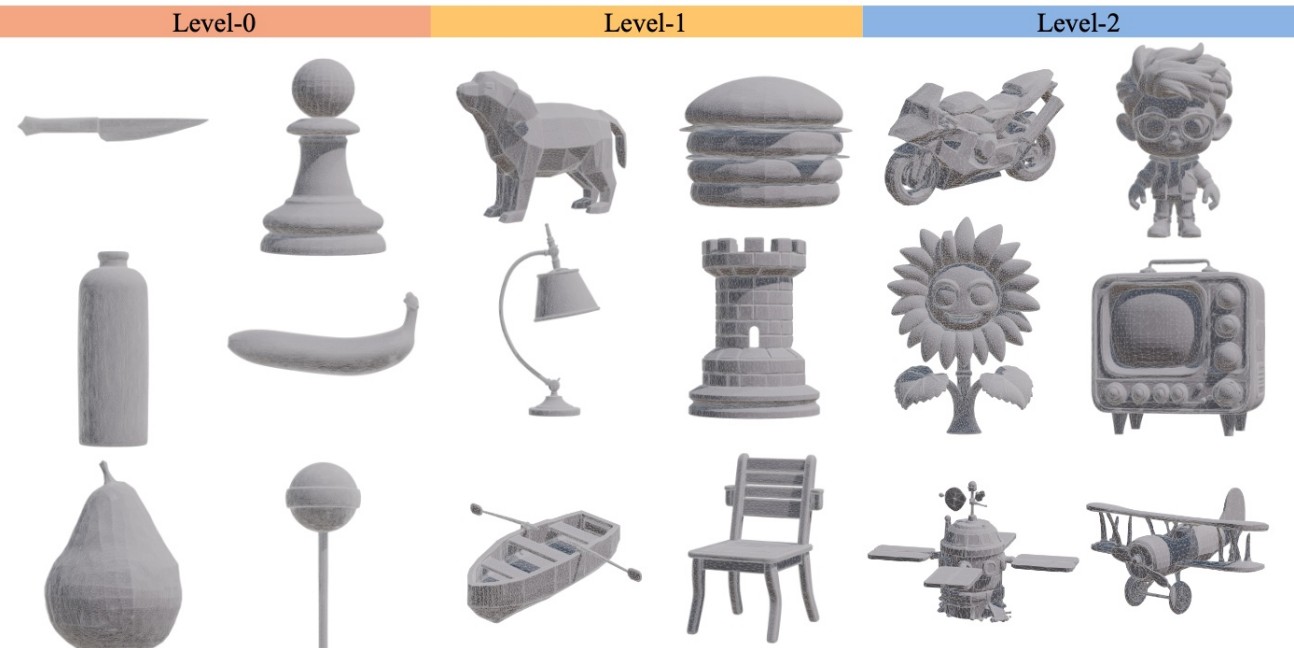

*Figure 7.* **A subset of examples from the test dataset.** Our test dataset contains a rich variety of shapes and is divided into three different difficulty levels: level-0, level-1, and level-2.

a truncated attention window and a maximum inference context size of 9,000 tokens—design decisions that result in fragmented meshes, as illustrated by the red boxes in Figure 4 of the main text. Furthermore, DeepMesh frequently produces meshes that are overly dense yet incomplete, as evidenced in rows 1 and 4. These shortcomings inflate its CD and HD metrics and diminish its user-study vote share.

In contrast, BPT omits any truncation window, resulting in more stable outputs and consistently robust performance across all test cases. The proposed XSpecMesh framework leverages BPT as its backbone: BPT's token sequences are employed to train cross-attention decoding heads, and each candidate token generated by these heads are subsequently verified by the backbone model. This pipeline ensures that the generated outputs closely match those of BPT in terms of CD and HD, and—given the perceptual indistinguishability—yields a user-study vote share effectively equivalent to BPT's. Finally, while preserving BPT-level quality, XSpecMesh achieves a $1.7\times$ speedup, thereby significantly reducing the backbone model's inference time.

## F. LoRA Instead of full Parameters Tuning

We fine-tune the backbone model using LoRA rather than full-parameter fine-tuning. Compared to full-parameter tuning, LoRA is more training-efficient and converges faster. Equally important, LoRA effectively prevents distribution drift in the backbone model. Since our method relies on the backbone to verify multiple candidate tokens, its predic-

tions are critical to generation quality. With full-parameter fine-tuning, gradients originating from the decoding heads can cause certain backbone parameters to drift significantly, harming sampling quality. By contrast, LoRA applies low-rank update matrices to the model; these low-rank updates curb any severe parameter drift induced by decoding-head gradients during training, thus preserving generation quality.

## G. More Results

We further collected more examples, and displayed the generated results of BPT and our method in Figures 8 and 9. In these challenging cases, our approach is capable of producing meshes with shape and topology quality comparable to that of the base model BPT, while significantly accelerating the generation speed.

## H. Broader Comparison

We provide a broader comparison with other autoregressive mesh generation methods in Table 5, including MeshSilk, MeshAnythingV2, and TreeMeshGPT. Among them, only MeshSilk achieves a better CD score than BPT, but its HD performance is worse than BPT and its generation latency is also higher. Therefore, considering both overall performance and inference efficiency, we believe BPT is currently the strongest method and is well suited as the backbone model for acceleration.

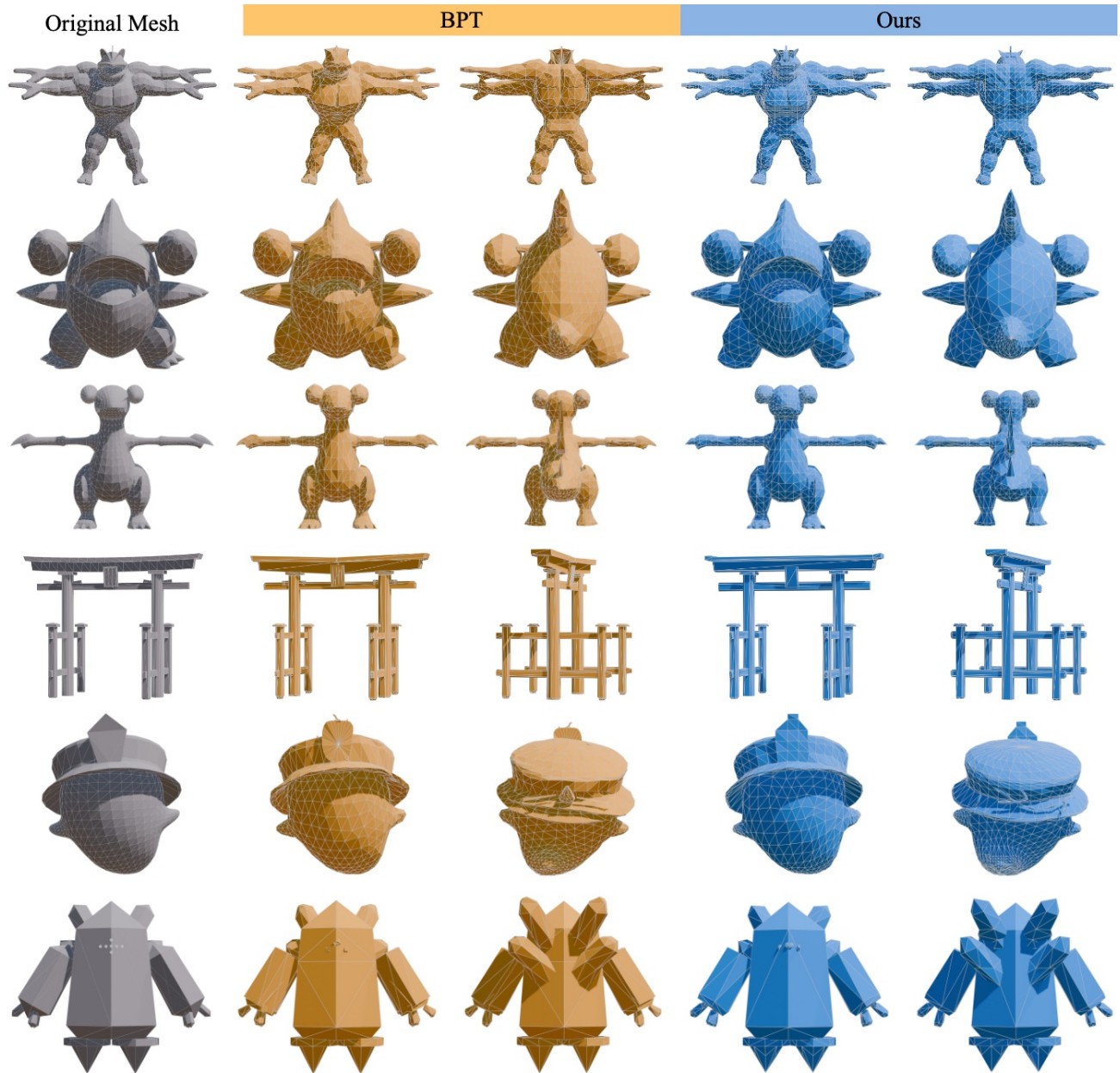

*Figure 8.* **Additional generation results of our method versus BPT.** Our acceleration method, built upon BPT, substantially accelerates generation while preserving BPT's shape and topological fidelity.

## I. Cross-Category Experiments

To further validate the generalization capability of our method, we conduct experiments on a cross-category dataset covering eight categories: monsters, humans, weapons, buildings, animals, plants, furniture, and decorative objects. The quantitative results are reported in Table 7. Our method performs well across categories, which can be attributed to the backbone model being extensively trained on a large-scale and diverse dataset, thereby endowing it with strong generalization ability that is not limited to specific cate-gories.

## J. Discussion

A concurrent work, FlashMesh(Shen et al., 2025a), also explores speculative decoding for accelerating mesh generation. Built on MeshXL's uncompressed mesh encoding and an hourglass Transformer architecture, FlashMesh proposes a hierarchical speculative decoding strategy. However, its design is closely tied to the underlying mesh serialization scheme, making it difficult to generalize to irregular

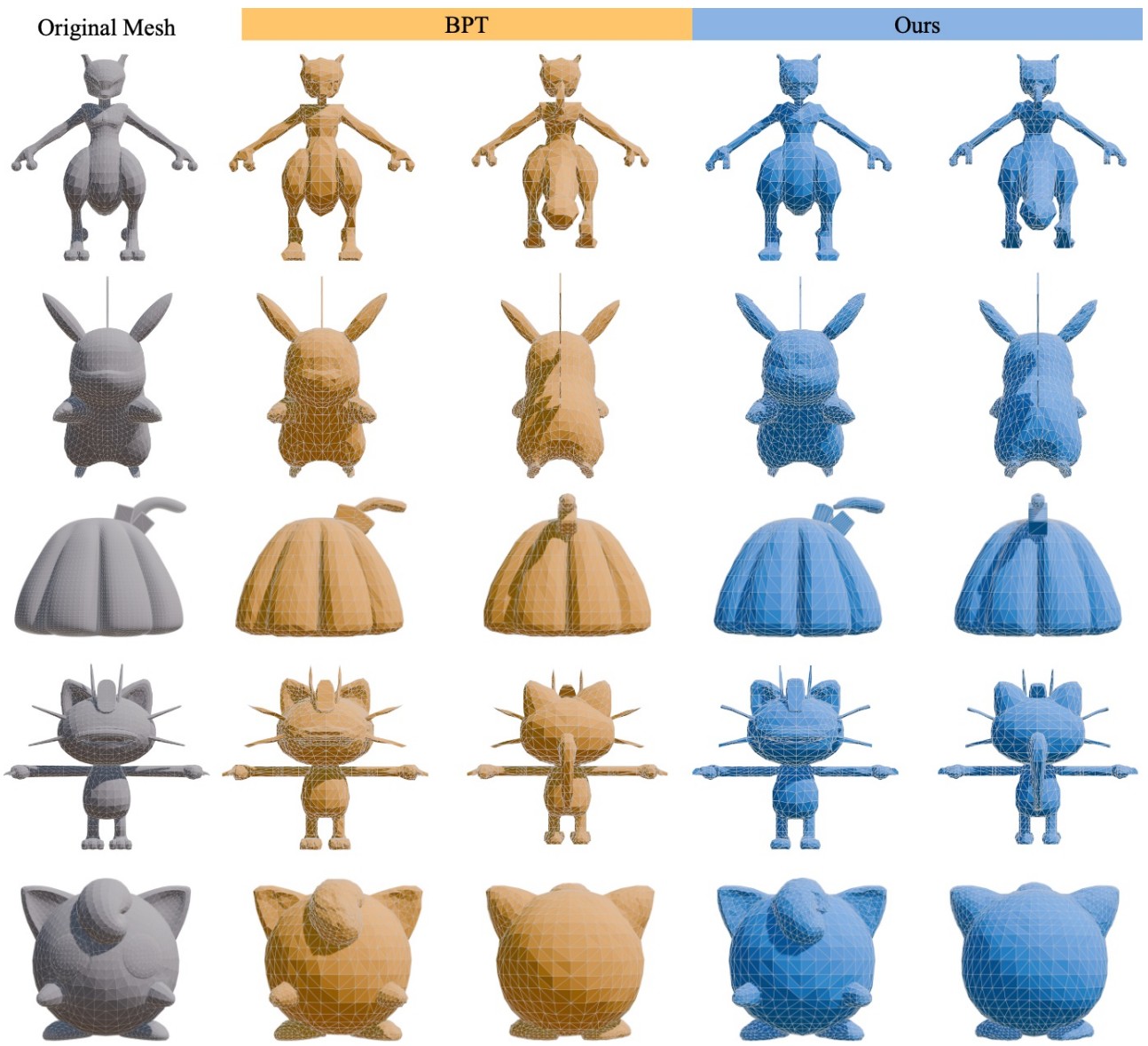

*Figure 9.* **Additional generation results of our method versus BPT.** Our acceleration method, built upon BPT, substantially accelerates generation while preserving BPT's shape and topological fidelity.

compressed encodings such as those used in BPT and Mesh-Silk. In contrast, we propose a speculative decoding method that better adapts to irregular mesh serialization. By using cross-attention decoding heads to learn irregular serialization patterns, our method enables more general acceleration for mesh generation.

## K. Generality

To further demonstrate the generality of our method, we also apply it to DeepMesh for acceleration. Using the same truncated-window training strategy as DeepMesh, the results are shown in Table 8.

Since DeepMesh adopts a hourglass transformer, its hierarchical structure introduces technical challenges for maintaining the KV cache. Specifically, the hourglass transformer performs hierarchical downsampling and upsampling over the sequence. We maintain a separate KV cache for each layer and record the valid KV cache positions (i.e., positions of tokens that have already been verified). Each time the KV cache is used, only the valid positions are accessed. When a token is verified, the valid positions in the KV cache are updated layer by layer. In addition, we employ hierarchical absolute position writes to the KV cache, meaning that at each forward pass, the newest values are written into the corresponding KV cache positions, overwriting previously

| Method | Watertight Ratio ↑ | Boundary Edge Ratio ↓ | Two-Manifold Ratio ↑ | Nonmanifold Edge Ratio ↓ |
|---|---|---|---|---|
| DeepMesh | 0.01 | 0.0569 | 0.01 | 0.0057 |
| BPT | 0.04 | 0.0362 | 0.04 | 0.0076 |
| Ours | 0.04 | 0.0320 | 0.04 | 0.0052 |
| MeshSilk | 0.00 | 0.0235 | 0.00 | 0.0002 |
| MeshAnythingV2 | 0.00 | 0.1328 | 0.00 | 0.0052 |
| TreeMeshGPT | 0.08 | 0.0613 | 0.08 | 0.0061 |

*Table 6.* **Evaluation metrics for watertightness and manifoldness.**

| Category | CD ↓ | HD ↓ |
|---|---|---|
| Monster | 0.1176 | 0.2110 |
| Human | 0.1045 | 0.2232 |
| Weapon | 0.0904 | 0.2108 |
| Building | 0.1232 | 0.2459 |
| Animal | 0.0969 | 0.2097 |
| Plant | 0.1191 | 0.2302 |
| Furniture | 0.1087 | 0.2268 |
| Decorative | 0.0914 | 0.2195 |

*Table 7.* **Category-wise CD and HD comparison.**

| Method | CD ↑ | HD ↓ | SCR ↑ | Speedup ↑ |
|---|---|---|---|---|
| MeshAnyV2 | 0.1708 | 0.3574 | 1.000 | 1.00× |
| +ours | 0.1767 | 0.3639 | 1.975 | 1.63× |
| MeshSilk | 0.1111 | 0.2293 | 1.000 | 1.00× |
| +ours | 0.1182 | 0.2385 | 1.828 | 1.55× |
| DeepMesh | 0.1323 | 0.2648 | 1.000 | 1.00× |
| +ours | 0.1378 | 0.2767 | 2.184 | 1.52× |
| BPT | 0.1165 | 0.2223 | 1.000 | 1.00× |
| +ours | 0.1168 | 0.2261 | 2.021 | 1.71× |

*Table 8.* **The acceleration results of applying our method to different foundation models.**

invalid entries.

## L. Analysis of Watertightness and Manifoldness

Watertightness and manifoldness are important indicators of mesh quality and play a crucial role in downstream applications such as physical simulation, 3D printing, rendering, and geometry processing. Table 6 reports four topology-related metrics, including watertight ratio, boundary edge ratio, two-manifold ratio, and nonmanifold edge ratio, for different methods, including DeepMesh (Zhao et al., 2025a), BPT (Weng et al., 2025), MeshSilk (Song et al., 2025), MeshAnything v2 (Chen et al., 2024c), TreeMeshGPT (Lionar et al., 2025), and our method. In general, higher watertight and two-manifold ratios indicate better topological soundness, while lower boundary edge and nonmanifold edge ratios indicate fewer structural defects in the generated meshes. The results show that autoregressive mesh generation remains challenging from a topological perspective. Most existing methods achieve relatively low watertight and two-manifold ratios, suggesting that guaranteeing strict mesh validity is still difficult at the current stage of autoregressive generation. At the same time, different methods exhibit different trade-offs: for example, TreeMeshGPT attains the highest watertight ratio and two-manifold ratio, while MeshSilk achieves the lowest boundary edge ratio and nonmanifold edge ratio. This indicates that current methods may be better at improving certain aspects of topology quality, but still struggle to consistently satisfy all desired mesh validity properties. Compared with the backbone model, our method maintains comparable performance on these topology-related metrics. Specifically, our method achieves the same watertight ratio and two-manifold ratio as BPT, while obtaining lower boundary edge ratio and nonmanifold edge ratio. Although our method does not achieve the best result on every individual metric, the overall results suggest that it preserves the topology quality of the backbone model while introducing inference acceleration. This observation further supports that our acceleration strategy does not significantly compromise mesh quality, even on challenging topological criteria such as watertightness and manifoldness.

