# OpenReview forum: "XSpecMesh: Quality-Preserving Auto-Regressive Mesh Generation Acceleration via Multi-Head Speculative Decoding"
_ICML.cc/2026/Conference — ICML 2026 regular_

### Official Review · Reviewer_sYrk · 2026-02-27

**Soundness:** 2
**Presentation:** 3
**Significance:** 2
**Originality:** 3
**Overall Recommendation:** 4
**Confidence:** 4

**Summary:**

This paper introduces XSpecMesh, an acceleration framework designed to address the high inference latency of auto-regressive mesh generation models. The method employs a lightweight, multi-head speculative decoding scheme that enables the parallel prediction of multiple tokens within a single forward pass. To maintain topological precision, a verification and resampling strategy is implemented where the backbone model filters and corrects sub-optimal predictions. Additionally, a distillation strategy is used to align the decoding heads with the backbone, improving inference efficiency without sacrificing mesh quality.

**Compliance With Llm Reviewing Policy:**

Affirmed.

**Final Justification:**

After discussion, although the paper still has some limitations, its strengths outweigh the weaknesses; therefore, I have raised my score.

**Key Questions For Authors:**

1. How well does the proposed method generalize? For example, is it sensitive to rotations of the point cloud? How does it perform when generating unseen objects within the same category?
2. How stable and robust is the method? If the same mesh is serialized using different ordering strategies to create separate training data, can the model still learn effectively and ultimately generate plausible and consistent meshes?

**Limitations:**

The paper lacks a discussion of the method’s performance with respect to generalization ability, training stability, and the watertightness of the generated results.

**Strengths And Weaknesses:**

Strengths:
1. The paper effectively boosts the inference efficiency of autoregressive models by implementing a multi-head architecture that parallelizes the prediction of multiple subsequent tokens, thereby mitigating the inherent bottlenecks of sequential next-token prediction.
2. The proposed method is supported by extensive ablation studies that rigorously validate the effectiveness of each core component.

Weaknesses:
1. The transition from sequential next-token prediction to parallel multi-token prediction is not sufficiently justified from a theoretical perspective. By predicting multiple tokens in parallel, the framework effectively decouples the causal dependencies where token $x_{t+1}$ typically conditions the subsequent token $x_{t+2}$. This relaxation of causal constraints increases the ill-posedness of the generation task. The authors should provide a formal analysis of how this parallelization affects prediction stability and error accumulation. A detailed discussion on the trade-offs between inference speedup and the increased entropy/complexity of the prediction space would greatly enhance the technical depth of the work.
2. The paper lacks quantitative metrics about the topological quality of the generated meshes. For 3D mesh generation, properties such as watertightness and manifoldness are critical for downstream applications. The authors should provide statistics on these kinds of metrics or explain how the model ensures the structural integrity of the output.
3. The manuscript lacks several critical details, making it difficult to assess the validity of the experimental results. Please clarify the serialization strategy, the composition of the test set, and how the input point clouds were generated.

---

> ### Author Rebuttal · Authors · 2026-03-29
>
> We appreciate your positive and constructive feedback, including comments such as “alleviating the inherent bottleneck of next-token prediction” and “rigorously validating the effectiveness of each component.”
>
> **W1. Causal constraints from next-token prediction to multi-token prediction**
>
> In fact, although we use **multi-token prediction**, we do **not** relax the dependencies am
> ong tokens in the unified sequence. This is because our **verification and resampling strategy** performs causal verification for each token generated in parallel through a causal mask. Only the tokens that pass verification are appended to the sequence, while those that fail verification are discarded directly. As a result, our generation quality remains comparable to that of the backbone model, enabling improved inference efficiency while preserving generation quality. Therefore, this parallelization does not harm prediction stability or introduce additional error accumulation.
>
> Regarding the trade-off between inference acceleration and the prediction space/complexity, this question can be answered from two perspectives:
>
> 1. **From the perspective of vocabulary prediction**, the vocabulary size of mesh generation models is usually fixed, and thus the complexity of the prediction space is typically also fixed.
> 2. **From the perspective of model structure**, as shown in **Figure 5**, we study the relationship between the number of prediction heads and the inference speedup ratio. The experiments demonstrate that **more heads are not always better**.
>
> **W2. Watertightness and manifoldness**
>
> We would like to clarify that, in the field of autoregressive mesh generation, metrics such as **watertightness** and **manifoldness** are usually not reported or compared \[1,2,3\].
>
> We train the model using data that contains extensive watertight processing and non-manifold structure handling. However, the autoregressive training paradigm may still generate non-watertight (broken) structures, as illustrated by the DeepMesh results in **Figure 4**. Methods that specifically enforce watertightness are typically designed in the **reinforcement learning** setting, whereas our method is a **pretraining-based acceleration approach**.
>
> **W3. Serialization strategy, test set composition, and how the input point clouds are generated**
>
> 1. **Serialization strategy:** We use the same serialization strategy as BPT, which compresses the sequence length through **block indexing** and **patchify** operations.
> 2. **Test set composition:** The composition of the test set is the same as in previous methods. We use image samples provided in the code repositories of Hunyuan3D 2.0 and Trellis as inputs, and generate 200 meshes as test examples to ensure that these data are unseen during training.
> 3. **Input point clouds:** The input point clouds are typically obtained either from scanned data or by sampling from the surfaces of meshes generated by diffusion-based models such as Trellis.
>
> We will provide the relevant details in the next version of the paper.
>
> **Q1. Generalization ability of the proposed method**
>
> 1. **Generality:** The proposed method has strong generalization ability.
> 2. **Sensitivity to point cloud rotation:** The method is not sensitive to the global rotation of the point cloud. In addition, because small Gaussian noise is added to the point cloud during training, the model exhibits good robustness at inference time.
> 3. **Generating unseen new objects within the same category:** Our method can not only handle unseen objects within the same category well, but can even generalize to new objects from different categories. In our evaluation experiments, these samples are not only from categories unseen by the model, but also exhibit a substantial domain gap from the training data, making them particularly challenging test cases.
>
> **Q2. Stability and robustness**
>
> 1. As discussed in **Q1**, the proposed method has good stability and robustness, and can stably generate new objects that the model has never seen before.
> 2. For the entire training set, all mesh samples must be serialized into one-dimensional sequences using a **consistent ordering strategy**, which is a necessary condition for autoregressive sequence generation. This has already been discussed in earlier works such as MeshGPT \[4\]. If multiple training samples are constructed from the same mesh using different ordering strategies, this would break the sequential consistency required by the autoregressive paradigm, making it difficult for the model to generate reasonable and consistent meshes.
>
> **References**
>
> \[1\] *Scaling mesh generation via compressive tokenization*[CVPR 2025].
>
> \[2\] *Deepmesh: Auto-regressive artist-mesh creation with reinforcement learning*[ICCV 2025].
>
> \[3\] *Mesh Silksong: Auto-Regressive Mesh Generation as Weaving Silk*[ICLR 2026].
>
> \[4\] *MeshGPT: Generating triangle meshes with decoder-only transformers[CVPR 2024]*.

---

> > ### Author Rebuttal · Reviewer_sYrk · 2026-04-02
> >
> > Thank you for the rebuttal. However, the following concerns remain unresolved.
> >
> > 1) Regarding the inherent next-token causal dependencies in mesh generation, the authors have not adequately addressed my question. They mention using a causal mask for token filtering, which indeed leverages dependencies between preceding and subsequent tokens. However, this only applies to filtering rather than the actual token generation process. While softening such dependencies enables parallel prediction, it also increases the complexity of the prediction space. The paper does not provide sufficient analysis or discussion on this trade-off.
> >
> > 2) Concerning topological quality evaluation, given that experimental results are already available, conducting such tests should not be difficult. This evaluation is necessary for a comprehensive assessment of the method’s performance.
> >
> > 3) With respect to experimental details, the presentation remains vague. A fair and transparent experimental setup is essential for properly evaluating the effectiveness of the method.
> >
> > 4) Regarding generalization, the authors claim that the method exhibits cross-category generalization. However, they neither provide theoretical analysis to justify this capability nor present empirical evidence to support the claim.

---

> > > ### Author Response · Authors · 2026-04-04
> > >
> > > 1.We would like to clarify that although the generation process weakens dependency relationships, our verification module enforces strong constraints to ensure that tokens added to the sequence conform to the dependency structure of the base model. This is because the verification process itself relies on the base model to infer, based on causal relationships, whether the next token is reasonable and meets the required criteria. If all tokens are rejected, the process degenerates into the base model’s standard next-token prediction. Therefore, our method still preserves the causal dependencies inherent in next-token prediction, which is also the reason we are able to achieve quality-preserving acceleration.
> > >
> > > 2.For topological quality evaluation, we have additionally incorporated relevant metrics such as watertightness and manifoldness, we have the same capabilities as the base model. What we would like to add is that, in the current field of autoregressive mesh generation, watertightness and manifoldness are not primary objectives; therefore, existing models generally perform poorly on these aspects.
> > >
> > > | Method | Watertight Ratio ↑ | Boundary Edge Ratio ↓ | Two-Manifold Ratio ↑ | Nonmanifold Edge Ratio ↓ |
> > > |-------------------|-------------------|-----------------------|----------------------|--------------------------|
> > > | DeepMesh | 0.01 | 0.0569 | 0.01 | 0.0057 |
> > > | BPT | 0.04 | 0.0362 | 0.04 | 0.0076 |
> > > | Ours | 0.04 | 0.0320 | 0.04 | 0.0052 |
> > > | MeshSilk | 0.00 | 0.0235 | 0.00 | 0.0002 |
> > > | MeshAnything v2 | 0.00 | 0.1328 | 0.00 | 0.0052 |
> > > | TreeMeshGPT | 0.08 | 0.0613 | 0.08 | 0.0061 |
> > >
> > > 3.Thank you for your suggestion. Here, we provide further clarification to supplement our rebuttal. Our evaluation protocol follows that of previous autoregressive mesh generation methods, including but not limited to BPT (CVPR 2025), DeepMesh (ICCV 2025), and MeshAnything v2 (ICCV 2025). Specifically, we first generate 3D shapes from image inputs using models such as Trellis and Hunyuan3D 2.0. We then extract surfaces from the 3D occupancy representations and sample point cloud coordinates along with their normals from these surfaces. These data are not only unseen by the model but also exhibit a significant domain gap with the training data, thereby forming a highly challenging test set. This test set construction protocol is consistent with that used in prior methods. We will include these experimental details in the next version.
> > >
> > > We believe that we have answered all the questions you asked about the experimental design. If any parts remain unclear, we would greatly appreciate your further questions or feedback.
> > >
> > > 4.We respectfully point out that our test set is entirely out-of-domain, demonstrating the model’s generalization capability. In addition, the test set is cross-category, covering a total of eight categories—monsters, human characters, weapons, buildings, animals, plants, furniture, and decorative objects—with 200 samples in total. We have also conducted separate experiments across different categories, and the evaluation metrics are as follows.
> > >
> > > | Category   |   CD   |   HD   |
> > > |------------|--------|--------|
> > > | Monster    | 0.1176 | 0.2110 |
> > > | Human      | 0.1045 | 0.2232 |
> > > | Weapon     | 0.0904 | 0.2108 |
> > > | Building   | 0.1232 | 0.2459 |
> > > | Animal     | 0.0969 | 0.2097 |
> > > | Plant      | 0.1191 | 0.2302 |
> > > | Furniture  | 0.1087 | 0.2268 |
> > > | Decorative | 0.0914 | 0.2195 |
> > >
> > > Due to the rebuttal format being limited to textual responses, we will include qualitative experiments across multiple categories in the next revision.
> > >
> > > We would appreciate it if you could review our responses and clarifications.
> > > If there are any remaining questions or concerns, we would be happy to further discuss and clarify.

---

### Official Review · Reviewer_XwQZ · 2026-03-12

**Soundness:** 3
**Presentation:** 3
**Significance:** 3
**Originality:** 2
**Overall Recommendation:** 4
**Confidence:** 4

**Summary:**

The paper proposes a multi-head speculative decoding based mesh generation method to accelerate the slow token-by-token face generation process. During the inference, the tokens generated can be wrong and the error can accumulate. To alleviate that, they have a verification and resampling approach with probability thresholding to make sure problematic tokens do not get in and are resampled. They also employ backbone distillation to align output distributions to backbone model. With these improvements they achieve 1.7x speed-up while preserving the quality.

**Compliance With Llm Reviewing Policy:**

Affirmed.

**Final Justification:**

I'm still at the positive side and favor acceptance. Although rebuttal provided good discussions, I have some concerns due to speed-up bottleneck.

**Key Questions For Authors:**

1) There are existing methods applying speculative decoding to mesh generation as described in weakness section. I expect a discussion to compare against them and make the difference against those methods clear. If that's describe well I'm willing to raise my score.
2) Why do you think such a speculative decoding mechanism only provides a speed-up of 1.7x. What do you think is bottlenecking the process? Some paragraph on that would be useful for the paper.

**Limitations:**

there's a limitation section in the appendix, but it makes sense if that section is presented in the main paper.

**Strengths And Weaknesses:**

Strengths:
* The mesh generation process is known to be slow due to required number of tokens that needs to be generated. The method approaches problem to come up with a speculative decoding method that has been used in LLM domain. I believe that's a promising direction in mesh domain and this paper comes up with a useful method to leverage speculative decoding.
* I liked the automatic token verification and refinement process to prevent bad tokens from accumulating errors.
* Good analysis on the selection of the parameters for verification criterion and number of decoding heads.

Weaknesses:
* Presentation:
    * Having different colored renderings for baselines and proposed methods makes it difficult to do proper evaluation. I suggest using uniform colors.
    * I don't fully understand why does the paper have Table 1 that compared vocabulary size of mesh generation methods and language models.
    * Table 2 doesn't highlight best performing method through bolding or underlinening the numbers.
* Originality:
    * There have been existing methods in the mesh generation domain working on speculative decoding such as FlashMesh (arXiv:2511.15618) and MeshPad (arXiv:2503.01425). These methods also claim to accelerate the whole mesh generation process using speculative decoding. It would be useful to discuss these methods and compare them against the current approach.

---

> ### Author Rebuttal · Authors · 2026-03-29
>
> We appreciate your positive and constructive feedback, including comments such as “token verification and correction prevent error accumulation” and “the detailed analysis validates the choice of verification parameters and the number of decoding heads.”
>
> **W1. Unified visual colors**
>
> Thank you very much for your suggestion. We will revise this in the next version.
>
> **W2. Comparison between mesh generation methods and language vocabulary size**
>
> What we would like to emphasize here is that, compared with autoregressive models for LLMs, autoregressive mesh generation models face more **region-specific challenges**. Specifically, LLMs usually have a much larger vocabulary, which makes decoding over the vocabulary more complex. As a result, speculative decoding for LLMs often relies on relatively complex draft models. In contrast, mesh generation models usually have a much smaller vocabulary, making vocabulary-level decoding simpler. Therefore, we specifically design a **multi-head decoding scheme** tailored to this setting.
>
> **W3. Highlighting the best method with boldface or underlining**
>
> Thank you very much for your suggestion. We will revise the presentation accordingly.
>
> **Q1. Differences from FlashMesh and MeshPad**
>
> FlashMesh proposes a hierarchical speculative decoding method based on the uncompressed encoding of MeshXL and an hourglass Transformer architecture. MeshPad, on the other hand, is based on the fact that each vertex is represented by three tokens corresponding to the \(x\), \(y\), and \(z\) coordinates, and designs a speculative decoding method that conditions on the \(x\)-coordinate and predicts the \(y\)- and \(z\)-coordinates.
>
> However, these designs depend heavily on the mesh serialization format and are therefore difficult to adapt to irregular compressed encodings such as those used in BPT and MeshSilk. In contrast, we propose a speculative decoding method that is better suited to irregular mesh serialization. By using a cross-attention decoding head to learn the underlying patterns of irregular mesh serialization, our method enables a more general acceleration framework for mesh generation.
>
> **Q2. Bottlenecks limiting the achievable speedup**
>
> We believe the limitations of speculative decoding speedup can be explained from the following three aspects:
>
> 1. Compared with large language models (e.g., \(7\)B or \(13\)B), autoregressive mesh generation models are much smaller in parameter size (around \(0.5\)B) and have much less parameter redundancy. This fundamentally limits the maximum achievable speedup.
> 2. Mesh sequences exhibit much stronger dependencies. In mesh serialization, a single vertex is usually serialized into multiple tokens, whereas in large language models, one token often corresponds to one or more characters. This makes the dependencies among mesh tokens much stronger, which in turn constrains the achievable speedup.
> 3. BPT and DeepMesh adopt compressed sequence representations with relatively low redundancy, which further limits the room for acceleration.
>
> We believe that the main bottleneck preventing further speedup lies in the parameter scale of the backbone model. Since existing mesh generation models are already relatively compact in size, achieving additional speedup is inherently difficult.

---

> > ### Author Rebuttal · Reviewer_XwQZ · 2026-04-03
> >
> > Thank you very much for the rebuttal and additional details provided. I've a few concerns remaining on how good the current method is compared to other speculative-decoding based methods. Authors provided conceptual difference against those methods, but I believe the paper would benefit from at least some runtime comparisons.
> >
> > In addition, authors listed reasons on why the bottleneck happens and I appreciate the analysis. It'd be nice to have a future work discussion on how to overcome the bottleneck. For instance, do we need better tokenizers?

---

> > > ### Author Response · Authors · 2026-04-04
> > >
> > > 1. Since FlashMesh has not yet been open-sourced, we only compared our method with MeshPad. Since MeshPad focuses on sketch-based mesh editing, while our task is point-cloud-based mesh generation, the two methods are designed for different tasks, making a fair quality comparison difficult. Therefore, we compared only their generation speed: MeshPad generates at 10.19 faces per second, whereas our method achieves 20.57 faces per second, demonstrating significantly higher generation efficiency.
> > >
> > > 2. From the perspective of speedup ratio, we believe that as the parameter size of the base model increases, parameter redundancy will also increase, and accordingly the speedup ratio of our method will improve. In terms of generation quality, as you mentioned, a better tokenizer could further enhance the quality of generation. Indeed, better tokenizers are currently one of the main research focuses in this field [1,2].
> > >
> > > [1] *Scaling mesh generation via compressive tokenization*[CVPR 2025].
> > >
> > > [2] *Mesh Silksong: Auto-Regressive Mesh Generation as Weaving Silk*[ICLR 2026].

---

### Official Review · Reviewer_qKU3 · 2026-03-13

**Soundness:** 2
**Presentation:** 2
**Significance:** 3
**Originality:** 3
**Overall Recommendation:** 4
**Confidence:** 3

**Summary:**

This paper presents XSpecMesh, a framework designed to address the slow inference speeds inherent in autoregressive 3D mesh generation models. By introducing a multi-head speculative decoding mechanism, the framework significantly accelerates generation by predicting multiple tokens in parallel. Combined with a verification and resampling strategy from the backbone network, XSpecMesh ensures that the final output quality remains lossless. Ultimately, this method achieves a 1.7× inference speedup without any compromise in generation quality.

**Compliance With Llm Reviewing Policy:**

Affirmed.

**Final Justification:**

The author's rebuttal addressed my concerns, primarily regarding the need for a more comprehensive experimental evaluation. I believe the current evaluation is now complete, and therefore, I have decided to increase my score to 4.

**Key Questions For Authors:**

Could the authors provide a comparison across a broader range of generative models? Specifically, I would like to see the differences in generation quality and speed compared to the models mentioned in the 'Weaknesses' section, as well as the performance of the proposed method when applied to these other architectures. If the authors can provide these experimental results and incorporate them into the revised version, I would be willing to raise my score.

**Limitations:**

Yes

**Strengths And Weaknesses:**

Strengths:
1. This paper addresses a significant challenge: accelerating generative inference. The proposed method is both practical and easy to implement, delivering impressive speedup results.
2. Notably, these substantial gains in inference speed are achieved with virtually no compromise in generation quality.

Weakness:
1. The experimental baselines for comparison are not yet comprehensive. I believe the following works should also be included for comparison in terms of both speed and generation quality: MeshAnything, MeshAnything V2, TreeMeshGPT, MeshSilk, FastMesh, etc.
2. The comparison in Figure 4 is not entirely fair. The author uses BPT as the base model for method validation. However, in terms of generation quality, the DeepMesh baseline itself exhibits poor mesh quality and is inferior to both BPT and BPT with XSpecMesh. It appears that DeepMesh does not serve as an effective baseline for comparison. Figure 4 only demonstrates that the proposed method does not cause significant degradation in the quality of models generated by BPT.
3. Based on points 1 and 2, the paper seems to lack effective comparative experiments; it is merely an ablation study of the proposed method on BPT, which is insufficient. This fails to prove that the method achieves SOTA generation quality (unless BPT is currently the top-performing method—which, to my knowledge, might be true, but it is not common knowledge and still requires comparative experiments for verification). Furthermore, it does not demonstrate that the proposed method has broad universality or effectiveness across other mainstream generative models.

---

> ### Author Rebuttal · Authors · 2026-03-29
>
> We appreciate your positive and constructive feedback, including comments such as “the method is practical and easy to implement” and “it achieves a significant inference speedup without degrading generation quality.”
>
> **W1.W2. The baseline comparison is not comprehensive enough**
>
> Thank you very much for your suggestion. We have further supplemented the experimental results with Meshsilk, Meshanythingv2, and Treemeshgpt. Due to the limited rebuttal time, we will include all results in the camera-ready version and provide further discussion.
>
> From the results below, only Meshsilk outperforms BPT on the CD metric, while its HD metric is worse than that of BPT, and its generation latency is also slower than BPT. Therefore, we believe that BPT is currently the best method in terms of overall performance and efficiency.
>
> |                | CD↓    | HD↓    |Avg lat.↓|
> |----------------|--------|--------|---------|
> | Deepmesh       | 0.1323 | 0.2648 |  979.6  |
> | BPT            | 0.1165 | **0.2223** |  257.6  |
> | Ours           | 0.1168 | 0.2261 |  151.4  |
> | Meshsilk       | **0.1111** | 0.2293 |  294.8  |
> | Meshanythingv2 | 0.1708 | 0.3574 |  **147.9**  |
> | Treemeshgpt    | 0.1871 | 0.3425 |  239.3  |
>
> **W3. More comprehensive comparisons, as well as broader generality and effectiveness**
>
> 1. As described in **W1** and **W2**, we have added more comprehensive experimental results to demonstrate that our method achieves optimal performance and generation efficiency.
> 2. **Broad generality and effectiveness.** We respectfully point out that we have included acceleration experiments on DeepMesh in the supplementary material, which demonstrate that our method is a general lossless acceleration paradigm.
>
> **Q1. Differences in generation quality and speed compared with other models, and the performance of the proposed method when applied to other architectures**
>
> As noted above, we have added a more comprehensive analysis of generation quality and speed compared with other models.
>
> Our method has strong generality. As long as the mesh generation model follows the autoregressive paradigm of next-token prediction, the relationships among irregular mesh sequence tokens can be learned through the cross-attention decoding head.
>
> In the supplementary material, we demonstrate the generality of our method through acceleration experiments on Deepmesh. Therefore, the proposed method can also be effectively applied to other methods (MeshSilk、MeshAnything V2), to achieve lossless acceleration. However, due to the rebuttal deadline, it is difficult to complete these experiments in time. We will present the experimental results of implementing our method on MeshSilk and MeshAnythingV2 in the next revision.

---

> > ### Author Rebuttal · Reviewer_qKU3 · 2026-04-02
> >
> > Thank you for your reply, which has solved my problem. I will raise my score.

---

> > > ### Author Response · Authors · 2026-04-03
> > >
> > > Thank you for your acknowledgment of our work and for raising the score. We truly appreciate your time and effort in reviewing our paper. We will carefully revise the manuscript according to your suggestions.

---

### Decision · Program_Chairs · 2026-04-30

**Decision:**

Accept (regular)

**Comment:**

After discussion and careful consideration of the reviews and rebuttal, the paper is accepted. Congratulations.

Reviewers agree that this work addresses the critical inference latency issue in autoregressive 3D mesh generation and proposes XSpecMesh, a quality-preserving acceleration framework based on multi-head speculative decoding. The rebuttal has thoroughly resolved major concerns.

All revisions from the rebuttal must be incorporated into the camera-ready version:
1. Supplement comprehensive comparisons with MeshSilk, MeshAnything V2, TreeMeshGPT, and MeshPad.
2. Clearly distinguish from FlashMesh and MeshPad and highlight generality to irregular mesh serialization.
3. Add detailed analysis of speedup bottlenecks and brief future directions.
4. Include watertightness, manifoldness, and other topological metrics in the main paper.
5. Explicitly clarify serialization strategy, test set composition, and point cloud generation pipeline.
6. Improve presentation: unify visualization colors, highlight optimal results in tables.
7. Add cross-category generalization experiments and analysis.
8. Move the limitation section from the appendix to the main paper.